# Extreme Overvalued Beliefs: How Violent Extremist Beliefs Become “Normalized”

**DOI:** 10.3390/bs8010010

**Published:** 2018-01-12

**Authors:** Tahir Rahman

**Affiliations:** Department of Psychiatry, Washington University in St. Louis, 660 S. Euclid, Campus Box 8134, St. Louis, MO 63110, USA; trahman@wustl.edu

**Keywords:** psychosis, delusion, overvalued idea, terrorism, mass shootings, violence, forensic psychiatry

## Abstract

Extreme overvalued beliefs (EOB) are rigidly held, non-deusional beliefs that are the motive behind most acts of terrorism and mass shootings. EOBs are differentiated from delusions and obsessions. The concept of an overvalued idea was first described by Wernicke and later applied to terrorism by McHugh. Our group of forensic psychiatrists (Rahman, Resnick, Harry) refined the definition as an aid in the differential diagnosis seen in acts of violence. The form and content of EOBs is discussed as well as group effects, conformity, and obedience to authority. Religious cults such as The People’s Temple, Heaven’s Gate, Aum Shinrikyo, and Islamic State (ISIS) and conspiracy beliefs such as assassinations, moon-hoax, and vaccine-induced autism beliefs are discussed using this construct. Finally, some concluding thoughts on countering violent extremism, including its online presence is discussed utilizing information learned from online eating disorders and consumer experience.

## 1. Introduction

Extreme overvalued beliefs are a predominant motive behind global and homegrown violent and terrorist attacks. This operational definition differentiates idiosyncratic, psychotic thinking from shared subcultural beliefs or ideologies [1,2]. Terrorism is a behavior that comes with enormous direct and indirect costs. Its detection and prevention are challenging due to the increasing presence of online and social media radicalization [3]. The term *extreme overvalued belief*, derived from earlier definitions, is critical to understanding the psychopathology seen in acts of terrorism, cults, mass suicides, mass shootings, and online radicalization. Media and behavioral health specialists often erroneously report terrorist attacks, mass shootings, and cults as based on delusional, paranoid, or obsessional beliefs. The definition of extreme overvalued belief was recently defined to aid in the development of a differential diagnosis during criminal responsibility evaluations [1,2]. The definition may have potential use as a tool in the primary prevention of violence stemming from extreme overvalued beliefs.

This article reviews the concept of an *overvalued idea*, first defined by Carl Wernicke and later applied to psychopathology seen in acts of terrorism, cults, and mass shootings [1,4]. The discussion will begin by describing how the term *overvalued idea* has evolved in the psychiatric literature in describing conditions such as eating disorders and its application to violent behavior seen in mass suicides, cults, terrorism, and online radicalization. Finally, the discussion will turn to how the social transmission of beliefs occur to gain a better understanding of how extreme overvalued beliefs form and thus has important implications for the prevention of terrorism and mass deaths stemming from extreme overvalued beliefs.

## 2. Background and Definition

There has previously been a lack of agreement as to what precisely constitutes an overvalued idea [5]. Clarification regarding the proper definitions of overvalued beliefs, delusion, and obsession is needed to understand the motive behind violent criminal acts [2]. Our group of forensic psychiatrists (Rahman, Resnick, Harry) at the *American Academy of Psychiatry and the Law* examined the motive in the Anders Breivik mass murder case [1]. In that case, Breivik held bizarre and rigidly held beliefs that were found to be motives behind his bombing and mass shooting attack, which resulted in the deaths of 72 mostly young people in Norway. A team of examining psychiatrists erroneously believed that he held delusions stemming from schizophrenia. After reviewing a second team’s findings that he was legally responsible for the crimes, the court declared that he, in fact, had extremist beliefs shared by other right-winged groups in Norway and not idiosyncratic, fixed, false beliefs from delusions. We introduced the term *extreme overvalued belief* to describe the rigidly held non-delusional beliefs that Breivik held during his attacks. We went further to apply the construct to explain motives behind other acts of terrorism such as the Oklahoma City bombing(McVeigh and Nichols) as well as the 9/11 terrorist attacks [1,2]. We defined the term *extreme overvalued belief* as follows:
An extreme overvalued belief is one that is shared by others in a person’s cultural, religious, or subcultural group. The belief is often relished, amplified, and defended by the possessor of the belief and should be differentiated from an obsession or a delusion. The belief grows more dominant over time, more refined and more resistant to challenge. The individual has an intense emotional commitment to the belief and may carry out violent behavior in its service.[1,2]

This definition expands on what was first described as an *overvalued idea* by Wernicke in *Gundriss der Psychiatrie* (1906) and later invoked by then psychiatrist-in-chief Paul McHugh at Johns Hopkins in response to the 9/11 terrorist attacks [4,6]. This definition may aid examiners in determining motives in violent attacks, where similar rigidly held, non-delusional beliefs are present. FBI forensic behavioral specialists Matt Logan and Reid Meloy have approached this topic with similar descriptive findings [7,8]. Logan discussed his findings in *Lone Wolf Killers: A Perspective on Overvalued Ideas*. He argues that lone wolf killers have over-idealizing beliefs that develop into such overriding importance that the beliefs define the identity of the individual [7]. Meloy, in response to the 9/11 attacks, submitted a description of a *Violent True Believer* to the Behavioral Analysis Program of the FBI. The purpose of the advisory paper was to gain psychological knowledge concerning the homicidal-suicidal state of mind seen in terrorism. The violent true believer, he argued, is an individual committed to an ideology or belief system that advances homicide and suicide as a legitimate means to further a particular goal [8]. These findings are congruous with extreme overvalued belief as the putative belief system and mechanism for the motives behind this type of violent behavior.

## 3. Ueberwerthige Idee (Overvalued Idea)

Carl Wernicke (1848–1905) was chair of psychiatry and neurology at Breslau and later chair at the University of Halle (Germany) in 1904, just before his premature death from a bicycle accident. He is honored by having his name describe neuropathology seen in Wernicke’s aphasia and Wernicke–Korsakoff syndrome, but his psychiatric contributions have largely been neglected [9]. *Gundriss der Psychiatrie* was recently translated from German into English. In the classical description of *Ueberwerthige Idee*, Wernicke stated that “special conditions must prevail before such overemphasis takes on an aberrant character. Normally there is contradictory evidence, which gradually corrects any overvaluation. For aberrant overvaluation, however, these counterarguments, demonstrably, are no longer accessible”. He characterized overvalued ideas as appearing completely normal. However, individuals may *acquire* an aberrant character and behavior [4]. Wernicke’s examples from the late 19th century to the early 20th century included people who committed suicide after the loss of a fortune, after being sentenced to dishonorable punishment, or after the death of a loved one [4]. Subsequent psychiatric literature has applied this definition to describe the psychopathology seen in other disorders with over-idealizing values including anorexia nervosa, body dysmorphic disorder, gender dysphoria, hypochondriasis, hoarding, morbid jealousy, litigious paranoid state, pseudocyesis, and social phobia [5]. Women that are unable to conceive may develop over-idealization about having a child to the point of committing suicide or even kidnapping and forcibly cutting out an infant from a pregnant woman’s womb [10].

Wernicke’s description also captures the beliefs described by FBI profilers as the motive of many mass shooters, terrorists, and assassins that have acted in an unpredictable and aberrant fashion [11]. Psychologists, economists, and others have not found a specific personality profile or situational condition (e.g., poverty, oppression, or lack of education) that explains terrorism. Such behavior does not always meet the classic personality disorder criteria seen in criminal psychopathology such as antisocial personality disorder. Instead individuals are described as having vague psychosocial problems in their background [12].

Two things have changed quite dramatically since Wernicke first described the overvalued idea almost a century ago: access to information (particularly online) and access to much more lethal types of weapons [13]. Applying Wernicke’s concept to the development of extreme overvalued beliefs, a pathway to violence can be drawn in the following steps: (1) there are a core set of beliefs normally shared by others in their culture/subculture; (2) as the individual is exposed to progressively more extremist information and perceives a lack of contradictory information, reinforcement and refinement of the extreme beliefs occur; (3) additional amplification is acquired and coupled with the use of harm to self and/ or others in its service.

## 4. DSM-5 Definition of Overvalued Idea

Psychiatry has historically differentiated obsessions, delusions, and overvalued ideas from each other. An obsessional belief is recognized by an individual as his own and intrudes forcibly into his mind. The belief is often unpleasant and efforts are made to resist it (ego-dystonic). A delusion is a fixed, false, and idiosyncratic belief (not shared by others). An overvalued idea, by contrast, is a preoccupying rigidly held belief that is shared by others in a person’s culture or subculture. The beliefs are not resisted as in obsessions, but instead are amplified and defended (ego-syntonic) [1,6]. Karl Jaspers believed that delusions were from a mental illness and that overvalued ideas stemmed from personality features [14]. Sims, interested in religious ideology, has argued that delusions are apparent in the context of other psychotic symptoms such as thought disorder, hallucinations, and functioning level. He also emphasized that, in psychotic individuals, the lifestyle, behavior, and direction of the personal goals of the individual subsequent to a religious experience are consistent with the natural history of a mental disorder rather than with a personally enriching life experience [1]. In other words, the simplest way to differentiate a delusion from a non-delusion would be to look for other evidence of a psychotic disorder. It is considered rare for an individual to have an isolated symptom of delusions in the absence of other psychotic symptoms. The prevalence of a delusional disorder, as defined in DSM-5, is 0.02%, versus 1.0% for schizophrenia. The prevalence of extreme overvalued ideas is currently unknown [2].

To confuse matters, the DSM-5 and DSM-IV have vague and difficult to trace definitions of overvalued idea. The manuals describe it as being “less than delusional intensity” and **not shared** by others in their cultural or subcultural group. The definition does not appear at all in any of the earlier DSM series. This places the current DSM-5 definition as the exact opposite of Wernicke’s definition, which states that it is a belief **shared** by others [4]. European psychiatrists have emphasized them in terms of categories with well-defined contours and discrete definitions. Although violence can stem from delusions seen in psychosis, clarification of definitions is needed to empirically study rigidly held, non-delusional, shared beliefs seen in cults, mass suicides, terrorism, and online radicalization [1,2,5]. Given this discrepancy, it is possible that individuals once categorized as having a “delusional disorder” may actually fit the Wernicke (not DSM) definition of overvalued idea, thus making isolated delusions an even more uncommon entity.

## 5. Extreme Overvalued Belief: Possible Etiologies

When seemingly normal individuals act in a bizarre manner, it is critical for psychiatrists to understand the cultural context of the behavior. Religious cults illustrate this concept well. The greatest episode of deliberate loss of American lives prior to 9/11 was the mass suicide of 909 inhabitants (304 children) of Jonestown, Guyana, from cyanide poisoning in 1978. Reverend Jim Jones, founder of the *People’s Temple* instructed his followers to commit suicide while spreading rigidly held beliefs of intelligence organizations conspiring against their temple [15]. Similarly, 39 followers of *Heaven’s Gate* died in a mass suicide in California in 1999. They held rigid non-delusional beliefs that their suicides would allow them to exit their “human vessels,” allowing their souls to go on a journey aboard a spaceship they believed to be following comet Hale–Bopp. Some men in the group underwent castration in preparation for their “afterlives.” The *Aum Shinrikyo* cult attacked a Tokyo subway with sarin nerve gas in 1995, killing nearly a dozen people and injuring nearly 5000 [16]. The *Movement for the Restoration of the Ten Commandments of God* was a “doomsday cult” religious movement in which 778 people perished in a series of poisonings and killings that were either a group suicide/mass murder by a group leaders after beliefs of an approaching apocalypse [17]. Parents of online recruited terrorists have reported that their children had been “brainwashed” after watching Islamic State cult (ISIS) propaganda videos. They held extreme overvalued beliefs that violent jihadist acts would allow them and their family to go to heaven [18,19].

When beliefs are shared by others, idiosyncratic beliefs can become normalized [20]. In a study of 343 psychiatric outpatients who described themselves as religious, participants were asked to give their view of a demonic causality of their illness. A high prevalence of such beliefs was not only found in schizophrenia (56%) but also in non-delusional patients: affective disorders (29%), anxiety disorders (48%), personality disorders (37%), and adjustment disorders (23%). The authors concluded that demonic influence should be interpreted against the cultural and religious background that is shaping causal models of mental distress in the individual [21]. Occasionally, rigidly held non-delusional beliefs can become amplified and overvalued by an individual or group leading to extreme behavior in its service.

Harvard psychiatrist Oliver Freudenreich’s *Guide to Psychotic Disorders* states that, “while most people would not jeopardize their careers or lives for an overvalued idea, some will (and are secretly regarded as heroes by those less inclined to fight for an idea)” [22]. It is possible that conforming to extreme beliefs occurs through time and group effects. Solomon Asch (1951) discovered the foundation for conformity studies. In his classic experiment, participants viewed a card with a line on it, followed by another with three lines labeled “A,” “B,” and “C.” One of these lines was the same length as that on the first card, and the other two lines were clearly longer or shorter. Participants were correct when relying on their own objectivity, but when placed in groups that deliberately mislead them, they wrongly interpreted the stimuli, believing that the majority must be correct, leading them to erroneously answer with the majority [23]. Subsequent studies have demonstrated that some people will passively agree and go along with the majority in a group for various reasons such as to avoid disharmony [24].

Studies also show that a majority of persons entering cults do not necessarily exhibit psychopathology and that current cult members appear psychologically well-adjusted, and generally demonstrate few conspicuous symptoms of psychopathology [25]. In 1983, the American Psychological Association created The Task Force on Deceptive and Indirect Techniques of Persuasion and Control to investigate whether “brainwashing” or coercive practices played a role in recruitment by new religious movements. The task force report was ultimately rejected because it “lacked scientific rigor” and the association decided that it did not have sufficient information available to take a position on this issue [26]. Perhaps the definition of extreme overvalued belief better operationalizes this concept of “brainwashing” seen in religious cult movements.

Although arguably different from religious cults, some individuals hold what appear to be odd or idiosyncratic beliefs such as conspiracy theories involving the perceived source of assassinations of leaders, extraterrestrial life, UFOs, etc. In some instances, individuals relish, amplify, and defend these beliefs to the point of harm towards themselves and others. Examples include the moon-landing hoax and vaccine-induced autism conspiracy theories [27]. American astronauts have been stalked by individuals claiming that the moon landing was faked by NASA. Astronaut Buzz Aldrin physically defended himself from a moon-hoax believer that was stalking him and attempting to force him to “swear on the Bible” that he walked on the moon [28]. Parental concerns about perceived vaccine safety issues, such as a purported association between vaccines and autism, though not supported by a credible body of scientific evidence, led increasing numbers of parents to refuse or delay vaccination for their children. Outbreaks of measles have sporadically been reported to be, in part, due to this disinformation. The most frequent reason for non-vaccination stated by 69% of the parents in one study, was concern that the vaccine might cause harm [29,30,31]. Overvalued ideas, many of which are transmitted online, are at the source of these types of socially transmitted behaviors and the consequences of these beliefs can be destructive and life threatening.

## 6. Psychopathology of Terrorism

Although there is general condemnation of acts of terror involving mass deaths, a better appreciation of its psychopathology may aid in its prevention. Psychologists have pondered this question in the past. Stanley Milgram, interested in the Jewish Holocaust, studied obedience to authority figures with a series of experiments. In his 1973 article, The Perils of Obedience, he stated that “ordinary people, simply doing their jobs, and without any particular hostility on their part, can become agents in a terrible destructive process. Moreover, even when the destructive effects of their work become patently clear, and they are asked to carry out actions incompatible with fundamental standards of morality, relatively few people have the resources needed to resist authority”. He went on to create an “agentic” theory in which the essence of obedience consists in the fact that a person comes to view themselves as the instrument for carrying out another person’s wishes, and they therefore no longer see themselves as responsible for their actions [32]. In the online world, authority figures can also be a powerful destructive force. Milgram’s experiment may explain why it is not uncommon for terrorists to view videos of charismatic figures online and subsequently carry out violent behavior in the service of extreme overvalued beliefs [33]. French psychologists Tarde and LeBon described “group mind” or “mob behavior” in the 19th century. This concept has been used by social scientists to describe lynching and how individuals may blindly follow the lead of others. Similar to the conformity experiments, “online crowds” have been described to gather virtually, behave and act collectively, and produce similar effects and phenomena [34].

Justifying their behavior through narratives from online and other propaganda sources, terrorists, mass shooters, and conspiracy believers possess a sense of moral superiority to justify their violent acts. Anders Breivik, the 9/11 attackers, the Unabomber (Ted Kacynski), the Oklahoma City bombers, and ISIS militants all possessed belief systems in which their respective view of moral superiority justified agentic action [33]. Extremist propaganda invokes narratives that utilize moral superiority in arguments calling for violence against innocents. They may argue that they are “freedom fighters” morally justified in killing the enemy. However, the taking of innocent, unarmed lives through violent means is an agreed upon, universally condemned act [35]. The United Nations Security Council has stated that any acts of terrorism are criminal and unjustifiable, regardless of their motivation, wherever, whenever, and by whomsoever committed [36]. Psychiatry traditionally holds acts of violence against oneself or others as being the threshold to define psychopathology [37].

Carl Jung defined archetypes that are in the “collective unconscious” of mankind. Collective unconscious refers to experiences shared by any race or culture. These can include themes such as love, religion, birth, death, and innocence. These experiences exist in the subconscious of every individual and are recreated in literature, cinema, theater, and art [38]. The taking of innocent life by an act of terror evokes the collective unconscious of all mankind. This archetype defines the fundamental framework for why acts of terrorism are never justified and, instead, are a form of psychopathology stemming from extreme overvalued beliefs.

## 7. Eating Disorders and Overvalued Ideas

In order to better understand Wernicke’s description of amplification of belief systems from overvalued ideas, it may be useful to examine the most widely studied type of behavior stemming from them—*eating disorders* [39]. The concept of an overvalued idea has been widely used to describe eating disorders, body dysmorphic disorders, and gender identity issues, particularly in the European literature [5]. The negative body image and cognitive distortions seen in eating disorders amplify in the mind of the individual, growing more dominant and form a powerful driving force behind starvation and excessive exercise, even to the point of death. Selective eating patterns also occur in which patients will limit their food intake to a narrow range of preferred foods. Attempts to widen their repertoire of foods are met with resistance and counter-arguments. The behavior is slow to develop and individuals do not suddenly develop eating disorders [39]. This parallels the restriction of ideology and amplification that is seen in violent behavior stemming from extreme overvalued beliefs [6].

## 8. Threat Assessment from Beliefs

The form and content of beliefs are important not only in psychiatric definitions, but also to determine risk of violence. Just as there is an increased risk of violence with certain types of delusions and not others, there is also an increased risk of violence with some types of overvalued ideas. Overvalued ideas involving body image and weight (eating disorders), an inability to conceive a child, or perceiving a defective body part (body dysmorphic disorder and gender dysphoria) may create self-destructive forces that can lead to suicide. By contrast, extreme overvalued beliefs and beliefs of moral superiority may create violent behavior. Similarly, delusions held by a person with schizophrenia that he is Jesus likely have a lower propensity towards violence than a person with persecutory delusions of being stalked by his neighbor [40]. The form and content of beliefs may also help explain why there is a much higher rate of male offenders seen in acts of terrorism or mass shootings and the predominately high female rate seen in anorexia nervosa. It is possible that both sexes experience a similar rate of overvalued ideology but have different types of behaviors stemming from the content of their rigidly held beliefs. Further studies are needed using new operational definitions to determine this.

## 9. Online Eating Disorder Data

Peer and media effects can contribute to the worsening of eating disorder behaviors by intensifying previously held rigid beliefs regarding weight and eating. A meta-analysis showed that pro-eating disorder websites have a large effect on refining and amplifying the dysfunctional body image and eating habits seen in eating disorders. In a Stanford cross-sectional study of 698 families of patients (aged 10–22 years) diagnosed with an eating disorder, 41% of patients visited pro-recovery sites, 35.5% visited pro-eating disorder sites, 25.0% visited both, and 48.7% visited neither. Of those that visited pro-eating disorder sites, 96.0% reported learning new weight loss or purging techniques. The authors concluded that pro-eating disorder website visits were prevalent among adolescents with eating disorders and that parents had little knowledge of this [41]. This parallels the reports from family members of terrorists and mass murderers that reported not knowing that they viewed radical Internet content prior to their violent acts.

Eating disorder treatment units are equipped with a comprehensive treatment strategy including group therapy that challenges patients’ dysfunctional narrative thereby extinguishing maladaptive behaviors. Research on the impact of censorship/criminalization of pro-eating disorder websites has concluded that it further isolates and insulates the eating disorder sufferer. Banning the use of these websites may in fact worsen the behavior—users migrate to other, more secret websites, and the overvalued ideas begin to hold more valence. Instead, balancing the pro-eating disorder content with pro-recovery content may be a more effective strategy. For example, in 2012, Instagram made some hashtags unsearchable and disabled accounts that promoted eating disorders. Studies found that pro-eating disorder hashtags actually multiplied as people instead used deliberately misspelled hashtags like #anorexique to circumvent banned terms. Instagram later released a tool where users could anonymously flag a photo they deemed concerning for an eating disorder. Instagram will send the user a note of support along with directions to resources where they can get help [42]. This may serve as an effective model for preventing the development of online extreme overvalued beliefs [12].

## 10. Countering Violent Extremism

Countering violent extremism (CVE) is a broad phrase that covers a wide array of approaches that have been advanced to reduce the radicalizing effects of extremist narratives. Under the Obama administration, the U.S. developed a Strategic Implementation Plan for Empowering Local Partners to Prevent Violent Extremism. It has three objectives: (1) enhancing federal community engagement efforts related to CVE, (2) developing greater government and law enforcement expertise for preventing violent extremism, and (3) countering violent extremist propaganda [43].

Several programs in Muslim states (Saudi Arabia, Iraq, Yemen, Indonesia, and Singapore) have utilized de-radicalization programs carried out by moderate Muslim clerics who engaged detainees in a theological dialogue about what they portrayed as a correct interpretation of Islam [44]. In response to growing right-winged extremism, the Canadian government has begun to consider similar counter-extremist narratives, building on the strengths and expertise of diverse sectors in society [45].

The Internet has been of increasing interest in efforts to counter violent extremism in recent years. Attempts have been made to classify and identify extremist social media messaging using linguistic features such as offensive terms to discriminate hate and extremism promoting messages from other messages [46]. To better understand how a radicalized individual begins to amplify his beliefs, an examination of the way the Internet utilizes individual consumer information for advertising is helpful. Digital advertisers have used small computer files called cookies and now use canvas fingerprinting to detect the subtle differences in the rendering of text to extract a consistent fingerprint that can easily be used without the consumer’s awareness. Publishers place paid advertising on their websites and mobile applications to provide customers with tailored products or services [47]. The information can help, for instance, to find more books by an author they enjoy reading or have a movie title suggested to them based on interest in a topic. The information is honed and tailored over time to accurately detect and influence shopping habits [48]. A similar feedback effect occurs with online information searches. Persons searching radical material on an online video may receive progressively more similar radical content with additional videos and social media newsfeeds [49].

The refinement of online eating disorder and consumer shopping consumption parallels extremist consumption of radical ideology. Coupled with authority figures, conformity, and group minds, a weapon is easily created online and spread virally. The number of views (“likes” or with emoticons) can also be fabricated to virtually enhance this powerful social effect [50].

## 11. Online Social Transmission of Extreme Overvalued Beliefs

Research has also shown that people on the political extremes are more likely to perceive large partisan differences and political polarization and to be intolerant of people with different political beliefs [51]. A study of 61 million Facebook users demonstrated that the online world can affect significant real-world behavior on a large scale. They revealed that the effect of social transmission on real-world voting was greater than the direct effect of the messages themselves, and nearly all transmission occurred between “close friends” who were more likely to have a face-to-face relationship. These results suggest that strong ties are instrumental for spreading both online and real-world behavior in human social networks. People are also reluctant to correct misinformation in their memories if it fits in with their political beliefs [52]. Some studies have offered solutions that can easily be adapted to online search engines and newsfeeds, such as providing people with a narrative that replaces the gap left by false information and focusing on facts rather than the myths [53]. Extreme and false beliefs are clearly a new cyber-weapon used by extremists and even by foreign powers during elections [54]. Further research may help elucidate the use of the Internet and search engine functions to identify and neutralize extremist and false narratives using what has been learned about online amplification and extreme overvalued beliefs from past online experience [55].

## 12. Conclusions

The available literature regarding the psychopathology of violent extremism will require new definitions and strategies to counter it. Ultimately, terrorism and mass shootings are behaviors—ones with a large magnitude of effects. Similar to psychiatry’s experience with eating disorders, it is likely that merely attempting to ban extreme/radical content would be ineffective and encourage further secrecy and extremism. Instead, providing viewers of extreme content with alternate material containing a contradictory, factual message may help to decrease the development of extreme overvalued beliefs. Social transmission (person to person) is a more effective tool and the production of counter-narrative messages could be created to serve several functions: (1) prevent the user from progressing to further extremist ideology through time (amplification of beliefs), (2) disrupt the extremist messaging through counter narratives, and (3) reduce the online crowd effect by modifying the number of extremist message “views.” A number of possibilities may be useful, and combining them with artificial intelligence may prove more effective. Advisory panels from different cultural, political, medical, ethical, and religious groups can provide needed counter-narratives. Large-scale public awareness can be used to interrupt the social transmission of psychopathology of overvaluation seen in mass murders, acts of terrorism, mass suicides, avoidance of immunizations, etc. Further public health initiatives must be created to prevent extremism. Public awareness of getting the facts and not discarding them in favor of false information or an overvalued ideology is important. Harnessing the power of the Internet to produce Internet tools for counter-narratives and automated fact checking and new artificial intelligence designs to chase and neutralize radical and incorrect facts may constitute useful research [56]. Consumer-driven Internet marketing may serve as important models in this endeavor. Education programs similar to ones used in eating disorders, tobacco products, driving while intoxicated, fire prevention, and seat belt utilization may also be helpful. All are now seen as important primary prevention programs. As weapons of mass destruction (biological, chemical, and nuclear) become more prevalent and easily accessible, it is critical that mankind begins the process of the primary prevention of violence stemming from extreme overvalued beliefs.

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
