# Peer review of "Extreme Overvalued Beliefs: How Violent Extremist Beliefs Become “Normalized”"

_behavsci, 2018, doi:10.3390/bs8010010_

Round 1

Reviewer 1 Report

The present manuscript examines extreme overvalued dysfunctional beliefs. I found the topic quite interesting and the opinions seemed helpful. The text provides a strong historical foundation. However, a mix of issues could be addressed to improve the paper.

The review of literature could be improved. i believe there are many other recent journal articles that address these topics. I would suggest moving away from online resources, blogs, and unpublished sources. I believe there are multiple typographical errors throughout the reference list.

The authors seem to focus on the social transmission of overvalued dysfunctional beliefs. I think this issue is important should become the central focus throughout the paper. Even the title would benefit from an expanded description of the topic being addressed in the manuscript.

The text is somewhat rambling and rather expansive in scope. i would find it helpful to organize according to several key subheading. Perhaps the authors could confront pathways of social transmission, providing subsections on religious groups, social media, online sources, and other main ways that deviant subgroups may find others who share their dysfunctional beliefs.

I would prefer to see an abstract that includes a concise summary of the main points addressed in the text. As it stands now, the abstract is brief and relies on empty preview statements.

I felt the conclusions section moved a bit too far from the material that has been reviewed in the text. I believe there is merit in these ideas, and it may be helpful to remain a bit closer to the core issues related to the social transmission of overvalued dysfunctional beliefs.

I hope these comments are using in sharpening the focus and clarifying the goals of the paper.

Author Response

Thank you for your review. 

We have extensively modified the manuscript.

We took the suggestion of expanding on the concept of social transmission of overvalued beliefs. The title was changed to reflect his, as you suggested. 

The paper's structure and better use of subheadings were done. 

The abstract will be changed. I cannot access it yet. 

warm regards- 

Tahir Rahman 

Reviewer 2 Report

The authors provide an interesting account for ‘extreme overvalued beliefs’ as a novel psychiatric construct, distinct from delusions and obsessions. The manuscript particularly focuses on this construct within a forensic context, as it may hold promise in accounting for violent behaviours, such as acts of terrorism. I think the topic is worthy of discussion, and I liked many of the points raised by the authors (e.g., using eating disorders as a point of comparison, particularly with regard to interventions that may be used to help people develop more balanced views). I also liked the return of the idea that such beliefs are typically ‘shared by others’ , which not only makes these beliefs distinct from ‘delusions’, but also has high relevance in this digital age, where one may find online groups/forums just about any ‘extreme belief’.

However, I feel the manuscript needs to be a little clearer in places. Despite stating multiple times that ‘extreme overvalued beliefs’ are distinct from delusions and obsessions, the authors do not seem to point out explicitly what these differences are. This seemed like a missed opportunity to shed some clarity on a topic that is seldom discussed. Moreover, I wondered if there are perhaps any shared aspects to these constructs – for example, do ‘extreme overvalued beliefs’ and ‘delusions’ share the same or similar psychological processes that might contribute to their formation or maintenance?

There were also a few sections where I wondered ‘how does this fit with the central argument, why is it relevant?’ – the manuscript would be improved by linking each paragraph back to the central point of defining what an ‘extreme overvalued belief’ is. For example, are the authors suggesting that ‘mass suicides, cults and obedience to authority’ examples of ‘extreme overvalued beliefs, or that they merely share the underlying processes? Given the topic is so rarely addressed, more clarity around what the construct is (and what it isn’t) is particularly warranted. I would also advise against using terms like ‘schizophrenic’ (line 205).

Author Response

Thank you for your review. It was very helpful. 

A major revision was completed as suggested. A better discussion of the definitions and evolving concept of overvalued idea was done. We were unable to find information as to how delusions and overvalued ideas are shared, instead we chose to keep them as separate entities. We linked back each section to the central theme of extreme overvalued beliefs as suggested. We removed the offensive term "schizophrenic."

Warm regards-

Tahir Rahman  

Round 2

Reviewer 1 Report

The authors have made extensive changes in their text. I am satisfied with the revised version.

Author Response

Thank you. 

Reviewer 2 Report

The authors have written a more focussed and coherent manuscript, with improved flow and a clearer message. I did wonder, however, if the key distinction between delusion and ‘extreme overvalued belief’ constructs whether the belief is idiosyncratic (delusion) or shared (overvalued)? If so, is a non-shared extreme overvalued belief a delusion – is that all that separates the constructs? This is how it reads. I also thought the revised title a bit too narrow, and suggests that the manuscript will outline the process by which ‘extreme beliefs’ become normalised. While this is what the manuscript does in part, it seems a missed opportunity not to use the name of the actual construct in the title, especially as this article serves an introduction to this construct.

Author Response

Thank you for the additional comments which were critical to understanding the EOB construct. I have added a new section based on this comment. Please see revised sections under subheading "DSM-5 definition of Overvalued Idea." The title was also given both a heading and subtitle as suggested. 

warm regards, 

Tahir Rahman